TECHNICAL RELEASE

# PhysiMeSS - a new physiCell addon for extracellular matrix modelling

Vincent Noël[1,2,3,*], Marco Ruscone[1,2,3,4], Robyn Shuttleworth[5] and Cicely K. Macnamara[6,*]

1 Institut Curie, Université PSL, F-75005, Paris, France
2 INSERM, U900, F-75005, Paris, France
3 Mines ParisTech, Université PSL, F-75005, Paris, France
4 Sorbonne Université, Collège Doctoral, F-75005, Paris, France
5 Altos Labs, Redwood City, CA, USA
6 School of Mathematics and Statistics, University of Glasgow, University Place, Glasgow, G12 8QQ, UK

## ABSTRACT

The extracellular matrix, composed of macromolecules like collagen fibres, provides structural support to cells and acts as a barrier that metastatic cells degrade to spread beyond the primary tumour. While agent-based frameworks, such as PhysiCell, can simulate the spatial dynamics of tumour evolution, they only implement cells as circles (2D) or spheres (3D). To model the extracellular matrix as a network of fibres, we require a new type of agent represented by line segments (2D) or cylinders (3D).

Here, we present PhysiMeSS, an addon of PhysiCell, introducing a new agent type to describe fibres and their physical interactions with cells and other fibres. PhysiMeSS implementation is available at https://github.com/PhysiMeSS/PhysiMeSS and in the official PhysiCell repository. We provide examples describing the possibilities of this framework. This tool may help tackle important biological questions, such as diseases linked to dysregulation of the extracellular matrix or the processes leading to cancer metastasis.

**Subjects** Software and Workflows, Bioinformatics, Structural Biology

**Submitted:** 02 July 2024

* Corresponding authors. E-mail: vincent.noel@curie.fr; Cicely.Macnamara@glasgow.ac.uk

Preprint submitted at https://doi.org/10.1101/2023.10.27.564365

Included in the series: ***PhysiCell Ecosystem*** (https://doi.org/10.46471/GIGABYTE_SERIES_0003)

## STATEMENT OF NEED

Agent-based models (ABMs) establish independent *agents* that each behave according to a specific set of rules. Due to the independence of each agent, an ABM allows for both a highly stochastic modelling environment and an extremely fine-grained investigation of processes. Specifically, an ABM allows us to consider physical phenomena such as the mechanical interactions between agents. Indeed, ABMs are a tool used to model biological systems on a refined scale, especially in comparison to continuum models, due to their focus on specific cell behaviours and not entire population dynamics. If the agents within an ABM are cells (or even sub-cellular components), then ABMs are capable of capturing detailed single-cell and inter-cellular processes. Individualising the cell behaviour can lead to emergent heterogeneity at the tissue scale, which may not be captured by a continuum model. Furthermore, ABMs may also incorporate sub-cellular processes that inform and drive cell behaviour [1, 2]. As such, ABMs offer a truly multiscale approach interlinking dynamics occurring between different scales, including sub-cellular, cellular- and tissue scales (Figure 1).

The extracellular matrix (ECM) is a highly complex structure that acts not only as a scaffold for cells and tissues but also as a platform through which cells can communicate.

**Figure 1.** Schematic of macro-micro scales, ranging from sub-cellular scale to tissue scale.

ECM is a key structure in many biological processes, such as embryonic development, tumour formation, and fibrosis. The ECM is comprised of a variety of secreted proteins that can vary depending on their location in the body. One of the main components of the ECM is collagen, a fibrous protein that gives the ECM its structure and rigidity, with collagen type I, the most abundant protein in the human body, constituting around 90% of all proteins [3]. Alongside the rigid collagen fibres is elastin, a thin fibre that provides the ECM with elasticity and the ability to be reshaped. Elastin is a dominant component in tissues that require a high degree of flexibility, such as the lungs, bladder, and skin.

Cells interact chemically with the ECM through the secretion of enzymes, namely a family of matrix-degrading enzymes (MDEs) called matrix metalloproteinases (MMPs). Although prevalent during cancer invasion, the secretion of MMPs is an essential cellular process that occurs throughout normal development and ageing. These enzymes can degrade components of the surrounding matrix, in particular collagen and elastin, with their breakdown resulting in the destabilisation of cross-links between collagen and elastin fibres and ultimately the loosening of the ECM structure. Certain cells (fibroblasts) are also able to create new ECM by excreting collagen and fibronectin. Creating new ECM forms the basis for the important healthy process of wound healing as well as pathological processes such as fibrosis. Cellular remodelling of the ECM through degradation, regeneration, and indeed reorientation allows cells control over their environment. Invasive cancer cells, in particular, can initiate the remodelling and reorientation of collagen fibres, providing tracks to facilitate the invasion process and thus favouring a more tumorigenic microenvironment [4].

Additionally, cells use the ECM as a means of mechanical communication or mechanotransduction. A cell can modify both its behaviour, such as its capacity for proliferation and migration, as well as its intracellular composition and chemical



concentrations, in response to signals from its environment. In particular, fibronectin (FN), a matrix protein that mediates cell–ECM interactions, binds to integrin receptors on the cell membrane. This reaction causes the FN-induced phosphorylation of fibroblast growth factor receptor-1 (FGFR1) which then preferentially activates the protein kinase AKT, resulting in FN-induced endothelial cell migration [5].

Reflecting on the mechanotransduction and the structure of the ECM, the membrane-bound mechanosensor Piezo1 is a stretch-activated ion channel that activates due to a change in the stiffness of the ECM. The activation of Piezo1 has been found to cause the cell to change the organisation of its cytoskeleton [6]. It has been further shown that the stiffening of the ECM and consequential activation of Piezo1 regulate cell numbers, showing that tissue stiffness is a key regulator of ageing in cells [7]. Tumours are often fibrotic due to the presence of cancer-associated fibroblasts (CAFs) that produce and deposit ECM. The classical puckering of breast cancer is caused by cancer invasion of the ligaments of Cooper (collagen fibres, which define the shape of the breast). Stiff fibrous tumours create the perfect environment promoted by cell–ECM interactions for cancer cell survival, invasion and metastasis, leading to particularly aggressive tumours. For a more in-depth review of the biochemical and biomechanical properties of the ECM, we refer the reader to excellent reviews elsewhere [8, 9].

The ECM and its interactions with cells have historically been simulated using many formalisms, such as continuum, agent-based, or mechanical models [10]. The open-source computational package PhysiCell [11], which implements a multiscale hybrid ABM (having both agent and continuum aspects), has hitherto been used to describe such processes as the invasion of cancer cells [12, 13], cancer-immune responses [14], and most recently the progression and spread of COVID-19 within the human body [15]. The underlying environment in which these cellular dynamics take place is typically modelled, in PhysiCell, as a distribution of some substrate of interest. The substrate can exist in the background of the tissue, potentially be secreted or taken up by cells, and can then in turn influence cellular behaviours. Often this influence causes cells to undergo some form of taxis, whereby the motility of cells is directed along the gradients of the substrate. Both haptotaxis (movement of cells up cellular-adhesion site gradients) [16] and durotaxis (movement of cells up gradients of matrix stiffness) [13] have been modelled in this way. Whilst such an approach is useful, it does not allow for fine-grained modelling of the specific extracellular matrix (ECM) components, which also, importantly, interact mechanically with cellular agents.

In this paper, we introduce a PhysiCell addon, PhysiMeSS (MicroEnvironment Structures Simulation), that allows the user to specify rod-shaped microenvironment elements such as the matrix fibres (e.g., collagen) of the ECM. This gives the PhysiCell user the ability to investigate fine-grained processes between cellular and fibrous ECM agents. However, we note that the applications of PhysiMeSS stretch beyond those wanting to model the ECM – as the new cylindrical/rod-shaped agents could be used to model, for example, blood vessel segments or obstacles within the domain. In the following section, we describe the implementation of PhysiMeSS in 2D. Most of what we document here is also appropriate for 3D, but some of the mechanical aspects are confined to 2D. We thus choose to focus on 2D for this article and the first release of PhysiMeSS, which was included in version 1.13.0 of PhysiCell. To describe how to use PhysiMeSS within PhysiCell, we provide simple examples that we describe in this article.



## IMPLEMENTATION

We choose to incorporate ECM agents, such as `PhysiMeSS_Fibre`, a derived class of the `Cell` class in PhysiCell, to inherit all the properties of cells while adding new components specific to matrix fibres. This construct allows us to utilise the pre-existing and future functionalities of PhysiCell. We also included `PhysiMeSS_Cell`, a new derived class of the `Cell` class, to add new properties to the cells that are specific to their interaction with matrix fibres. Note that both of these new classes do not inherit directly from `Cell` but indirectly through `PhysiMeSS_Agent`, which contains methods specific to the PhysiMeSS addon and common between `PhysiMeSS_Fibre` and `PhysiMeSS_Cell`. More details about the implementation of these new agents can be found in Appendix A. This new implementation can be found within the addons/PhysiMeSS directory of PhysiCell, where additional documentation and user guidance are also provided.

PhysiMeSS comes with a dedicated sample project `physimess-sample` that can be built, as with all other PhysiCell sample projects, by typing `make physimess-sample; make` from the PhysiCell root directory. Once built, PhysiMeSS loads its own specific `config` and `custom_modules` directories along with `Makefile` and `main.cpp`. The custom PhysiMeSS set-up is found within `custom_modules/custom.cpp` and user examples are found within the `config` directory. We will first discuss fibre initialisation with simple examples provided in `config/Fibre_Initialisation`.

### Initialising cylindrical matrix fibres

The first key component of the PhysiMeSS tool is to allow the user to distinguish (visually and practically) between cellular agents (typically circular in 2D or spherical in 3D) and the fibrous rods of the ECM. These new agents are cylindrical, so they are designed to model matrix protein fibres (such as collagen) but could model any other cylindrical-shaped component the user requires (e.g., blood vessel segments). For PhysiMeSS to identify agents as cylindrical matrix fibers, the user must, for now, give the agent a name containing any of the following recognised strings: {*ecm, fiber, fibre, matrix, rod*}. The initialisation of cylindrical matrix fibres is incorporated in the `setup_tissue` function in `custom.cpp`. Using the typical centre-based approach, as implemented in PhysiCell, fibres are located within the domain using the position of their centre. The fibre centres, as per cell centres, can be read from a .csv file or can be assigned randomly across the domain. The number of initial fibres can be controlled distinctly from the number of cells, as a part of the user parameters in the .xml file. Further to the fibre centre, fibres are described using a length (which may be normally distributed), radius (which may be normally distributed) and an orientation angle (which may be normally distributed). These are controlled by custom data expressed in the .xml file, specific to each fibre definition (thus allowing different types of fibres) and detailed in Table 1. For further details on the fibre setup, see Appendix B.

#### *Visualising cylindrical matrix fibres*

Visualising cylindrical agents is not a simple task. However, since matrix fibres are typically much longer than their girth (for example, collagen fibres average 75 μm in length but only 2 μm in radius), we make the choice to visualise the cylinders as lines, at least for this release. We added two functions in the `PhysiMeSS` addon: `fibre_agent_SVG` and `fibre_agent_legend`. These functions are replacing PhysiCell's default drawing function and instead draw ECM agents as lines in the outputted SVG plots and legend. In addition, once

**Table 1.** User parameter values used for the initialisation of fibres read from .xml file.

| Parameter | Description |
|---|---|
| `fibre_length` | Average length of fibres. |
| `length_normdist_sd` | Standard deviation of fibre length allowing fibres to be prescribed a length from a normal distribution with mean length `fibre_length`. If set to zero, all fibres have the same exact length. |
| `fibre_radius` | Average radius of fibres. |
| `length_normdist_sd` | Standard deviation of fibre radius allowing fibres to be prescribed a radius from a normal distribution with mean radius `fibre_radius`. If set to zero, all fibres have the same exact radius. |
| `anisotropic_fibres` | Flag for fibre anisotropy. Isotropic fibres (if 1) will have heterogeneous randomly assigned orientations. Anisotropic fibres (if 0) will have a prescribed orientation based on `fibre_angle` and `angle_normdist_sd`. |
| `fibre_angle` | Average orientation angle of fibres (if anisotropic). |
| `angle_normdist_sd` | Standard deviation of fibre orientation angle allowing fibres to be prescribed an orientation angle from a normal distribution with mean angle `fibre_angle` (if anisotropic). If set to zero, all fibres will be aligned in the same direction. |

fibre cross-links have been calculated (after at least one mechanics timestep), the function `fibre_agent_SVG` colours the fibres according to the number of cross-links (more details in the section *Determining fibre cross-links)*.

In Figure 2 we display the results of three simple 2D fibre arrangements, which can be reproduced using the .xml settings file `Fibre_Initialisation/mymodel_initialisation.xml`. In all three cases, we attempt to initialise 2,000 fibres of length 75 μm and radius 2 μm randomly (there is no .csv file prescribing their centres) across an 800 μm × 800 μm domain. In the left panel, fibres are placed isotropically (`anisotropic_fibres` is false) and so are given randomly assigned orientations. In the middle panel, fibres are placed anisotropically (`anisotropic_fibres` is true) with a prescribed orientation angle of 0.2 radians, so all fibres are aligned at an angle of 0.2 radians from the horizontal line. In the right panel, fibres are placed anisotropically with a prescribed orientation angle distributed around a mean of 0.2 radians with a standard deviation of 0.15 radians. We note that during the fibre setup, if a fibre is not contained wholly within the domain, it is disregarded. If the fibrous mesh is isotropic, we attempt to re-initialise such disregarded fibres, up to a maximum of ten times, by giving them a new orientation within the domain. If, after which time, the fibre still does not sit wholly within the domain, or the fibrous mesh is anisotropic (in which case changing the orientation will not be possible), it is permanently disregarded. This is why, in all three cases shown in Figure 2, the actual number of initialised fibres is less than 2,000. We note that the algorithm that disregards fibres not wholly contained within the domain leads to differences in fibre density at the boundary and may cause preferred fibre alignment with the boundary. For the cases we consider, we are interested in behaviours in the centre of the domain and far from the boundary; thus, this does not matter. Should the reader wish to study behaviours close to the boundary, they may wish to modify this algorithm or indeed use their own customised algorithm for fibre initialisation.

## Fibre specific geometric considerations
When modelling fibres as cylinders there are certain geometrical aspects to overcome. These are discussed in the following Sections.

### Determining neighbours of cylindrical agents
One immediate aspect to overcome in modelling cylindrical agents is how to determine neighbours. Finding neighbours is important in ensuring the efficiency and effectiveness of

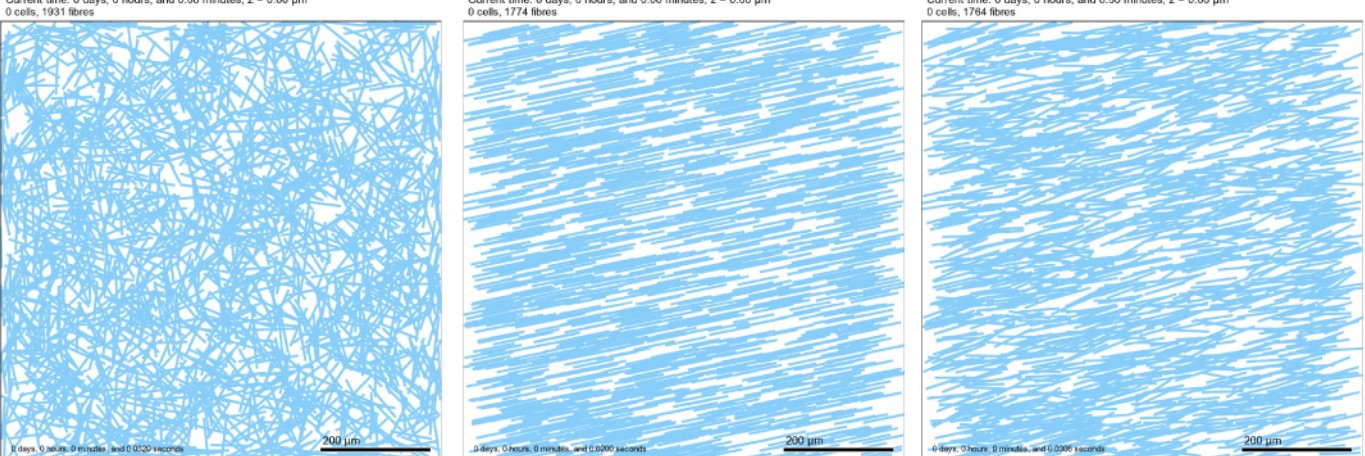

**Figure 2.** 2D Tissue domain with fibres (visualised as light blue lines) placed isotropically (left), anisotropically with an angle of 0.2 radians and standard deviation of 0.0 (middle) and anisotropically with an angle of 0.2 radians and standard deviation of 0.15 (right). Note: the time stated at the top of the figure is the simulated time and the time stated in the bottom right is the time taken to run the simulation.

the code; for example, mechanical interactions only occur between neighbours. Currently, cellular agents in PhysiCell belong to a single *voxel,* and neighbours are found by checking for agents in the same or adjacent *voxels*. Fibres are typically much longer than a typical cell diameter (for example, collagen fibres are 75 μm in length; cells are 10 μm in diameter), so it is likely that they will extend across multiple *voxels*. Thus, to find neighbours of fibres, we need to correctly find and register each *voxel* a fibre passes through. To achieve this, we implemented custom methods to find neighbours in PhysiMeSS. Since these methods are common to both cells and fibres in PhysiMeSS, they are defined in the `PhysiMeSS_Agent` class. The method for finding and registering a fibre to its *voxels* is `register_fibre_voxels`, which is implemented differently for cells and fibres. We then build our own list of neighbours by first finding all the *voxels* that an agent, cell or fibre sits within, as well as all adjacent *voxels*. Neighbours are then determined to be any agents that are registered to any of these *voxels* via the method `find_agent_neighbors`. After finding neighbours, we de-register the fibre from all *voxels* bar the *voxel* containing its centre using the method `deregister_fibre_voxels`. Further details of these processes are provided in Appendix C.

### *Determining fibre cross-links*

A key aspect of the ECM is that fibres cross-link to form the matrix network. Within PhysiMeSS, we consider, at a basic level, that two fibres cross-link at any point that they touch or intersect. Within the `PhysiMeSS` class, we determine whether a pair of fibres cross-link using `PhysiMeSS_Fibre`'s method `check_fibre_crosslinks`. This is primarily a geometrical problem and is explained in detail in Appendix C. It is only necessary to check cross-links for pairs of fibres that are in close proximity; thus, `PhysiMeSS_Fibre`'s method `add_crosslinks` determines cross-links between neighbouring fibres by calling the function `check_fibre_crosslinks` only if a pair of fibres are neighbours. Once fibre cross-links have been calculated, we colour the fibres according to the number of cross-links. This colouring follows a gradient of blue, with non-cross-linked fibres coloured light sky blue, fibres cross-linked once coloured steel blue, fibres cross-linked twice coloured blue and fibres

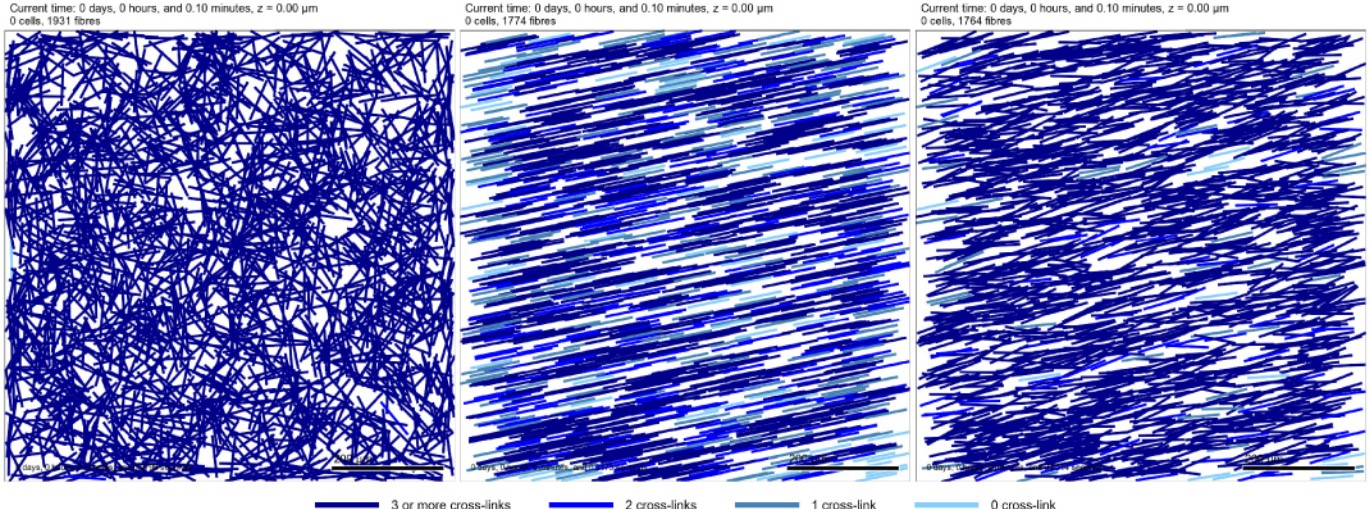

**Figure 3.** 2D Tissue domain with fibres as per Figure 2 but now coloured depending on the number of cross-links. Fibres with no cross-links are coloured light sky blue, fibres cross-linked once are coloured steel blue, fibres cross-linked twice are coloured blue and fibres cross-linked three or more times are coloured dark blue. Note: the time stated at the top of the figure is the simulated time and the time stated in the bottom right is the time taken to run the simulation.

cross-linked three or more times coloured dark blue. This is observed in Figure 3 where the initialised ECM fibres, as in Figure 2, are now coloured according to the number of cross-links. As an aside, and not unsurprisingly, we note an increase in the number of crosslinks (more dark blue fibres) for ECMs with more anisotropy.

### *Determining the nearest point on a fibre*

A further geometrical problem is determining the nearest point on a fibre from any other given point, or more precisely, the displacement vector from a fibre to a point. This is calculated in `PhysiMeSS_Fibre`'s method `nearest_point_on_fibre` and is used within the `check_fibre_crosslinks` method. It is also used to determine the nearest point from a cell centre to a neighbouring fibre which we use to inform mechanical and chemical interactions between cells and fibres, see the section *Fibre Specific Mechanics*. Determining the nearest point on a fibre is discussed in more detail in Appendix C.

### Fibre specific mechanics

The main purpose of introducing ECM agents as specific stand-alone agents rather than modelling them as a substrate or density is that we can more closely model the interaction between cells and ECM agents. For this paper and the associated release of PhysiMeSS, we focus on the implementation of mechanical interactions. Mechanical interactions are handled in PhysiMeSS (in an analogous way to PhysiCell) by potential functions that affect an agent's velocity. Thus, PhysiMeSS has a modified function `physimess_update_cell_velocity` that calls different potential functions as detailed in the following section. We note that PhysiMeSS has been written with some basic simplified mechanical interactions as standard, but the user is free to incorporate their own. Details of how the user can build in custom mechanical functions are included in Appendix D.

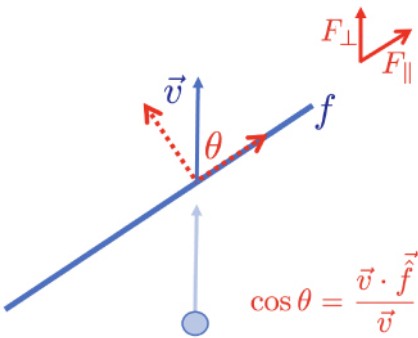

**Figure 4.** Schematic showing the adhesive and repulsive forces for a cell with velocity $\vec{v}$ interacting with a fibre, $f$, where $\hat{\vec{f}}$ is the unit vector in the direction of $f$.

### Potential functions

There are four possible interactions to consider, namely, cell–cell interactions, cell–fibre interactions, fibre–cell interactions and fibre–fibre interactions, thus four different potential functions are considered. Cell-cell interactions are handled by PhysiCell as per the function `add_potentials`. We do not go into detail here; more details can be found in [11].

Cell-fibre interactions are cell interactions with neighbouring fibres which affect the velocity of the cell, i.e. describe how a fibre affects a cell. These interactions are contained within the method `add_potentials_from_fibre` of the `PhysiMeSS_Cell` class. Firstly, we include simple repulsion and adhesion between cells and fibres as per between cells, copying the behaviour in `add_potentials`. This prevents cells from being able to go through fibres. Secondly, we introduce fibre-directed movement of cells as per [17] since cells are known to track along matrix fibres [18, 19]. We describe this briefly here, for more details please see [17]. We assume that cells move in response to a close neighbour fibre in two directions; an additional adhesive force, parallel to fibre orientation and an additional repulsive force orthogonal to the fibre (see, e.g., [20]) allowing the cell to track along the fibre.

The additional adhesive force takes the form

$$\mathbf{F}_{\parallel} = \alpha_{\text{fibre}} \left( 1 - \frac{|\vec{v}|}{v_{\max}} \right) \left( \frac{\left| \vec{v} \cdot \hat{\vec{f}} \right|}{|\vec{v}|} \right)^{s} \hat{\vec{f}}. \tag{1}$$

It is directed along the normalised direction of the fibre, $\hat{\vec{f}}$, and depends on the normalised scalar product between $\hat{\vec{f}}$ and $\vec{v}$, the velocity of the cell, i.e., proportional to the vector projection of $\vec{v}$ onto $\hat{\vec{f}}$ (Figure 4). Moreover, the force depends on an adhesion coefficient, $\alpha_{\text{fibre}}$, and on a threshold velocity, $v_{\max}$, which limits the pulling effect of fibres. We note that this equation implies the need for $|\vec{v}| \leq v_{\max}$. In the PhysiMeSS code, these parameters are called `vel_adhesion` and `cell_velocity_max`, respectively. The additional parameter $s > 0$ can be used to model additional effects that might increase ($s < 1$) or decrease ($s > 1$) the pulling effect. Following [17], we use $s = 1$.

The additional repulsion force is modelled as friction exerted by the fibre and takes the form

$$\mathbf{F}_{\perp} = \beta_{\text{fibre}} \left( \frac{|\vec{v}|^2 - |\vec{v} \cdot \vec{f}|^2}{|\vec{v}|^2} \right)^{r/2} \vec{v}. \tag{2}$$

It is directed parallel to the cell's current velocity, **v**, and is affected by the component of cell velocity orthogonal to the fibre, i.e., proportional to the vector rejection of $\vec{v}$ onto $\vec{f}$ (Figure 4). The strength of this force is dictated by $\beta_{\text{fibre}}$, a cell–fibre friction coefficient, and the exponent $r > 0$ can be used to model nonlinear effects that increase ($r < 1$) or decrease ($r > 1$) the repulsion forces. Following, [17] we use $r = 2$. In the PhysiMeSS code, this parameter is called `vel_contact`. The additional cell–fibre interaction force is computed as the difference of these adhesion and repulsion terms, $\mathbf{F} = F_{\parallel} - F_{\perp}$.

Fibre-cell interactions are fibre interactions with neighbouring cells that affect the fibre. These interactions are contained within `PhysiMeSS_Fibre`'s method `add_potentials_from_cell`. Any fibre–cell interaction can only take place if a fibre has no more than one cross-link with another fibre; otherwise, we assume the fibre is fixed in place, tethered by its cross-links.

A single fibre with no cross-links can be pushed and/or rotated by a motile cell. Whether pushing or rotation are turned on can be controlled by the user using the Boolean flags `fibre_pushing` and `fibre_rotation`. Cell-fibre pushing provides repulsion to the fibre, giving it speed and moving its centre from its initial position. Cell-fibre rotation maintains the position of the fibre centre and modifies the coordinates of its extremities by changing the fibre orientation. The new fibre orientation is based on moment arm magnitude, impulse, and fibre length. The moment arm magnitude is determined using the Euclidean distance between the origin and the point of impact. The impulse is obtained by considering the friction of the fibre (modified by the user with parameter `fibre_sticky`) and the speed of the interacting cell. The calculated impulse and fibre length are then used to determine the angular velocity, representing the rate of change of the fibre's orientation. Finally, the new orientation is obtained by rotating the old orientation vector using trigonometric functions. As a result, a cell impacting on a fibre end far from the centre will change the fibre's orientation faster than impacting closer to the centre.

The mathematical equations that correspond to code implemented to reproduce the fibre rotation upon contact with a cell, are the following:

**Moment arm magnitude**:

$$M = \sqrt{p_x^2 + p_y^2}. \tag{3}$$

- $M$: Moment arm magnitude. $p_x$, $p_y$: x and y components of the point of impact. This equation calculates the perpendicular distance from the fiber's axis to the force application point.

**Impulse**:

$$J = k \cdot v \cdot M. \tag{4}$$

- $J$: Impulse. $k$: 'fibre_sticky', a constant modifying the impulse. $v$: Cell's migration speed. $M$: Moment arm magnitude. This equation determines the impulse applied to the fibre, factoring in the cell's speed and the distance from the point of impact to the fibre's axis.



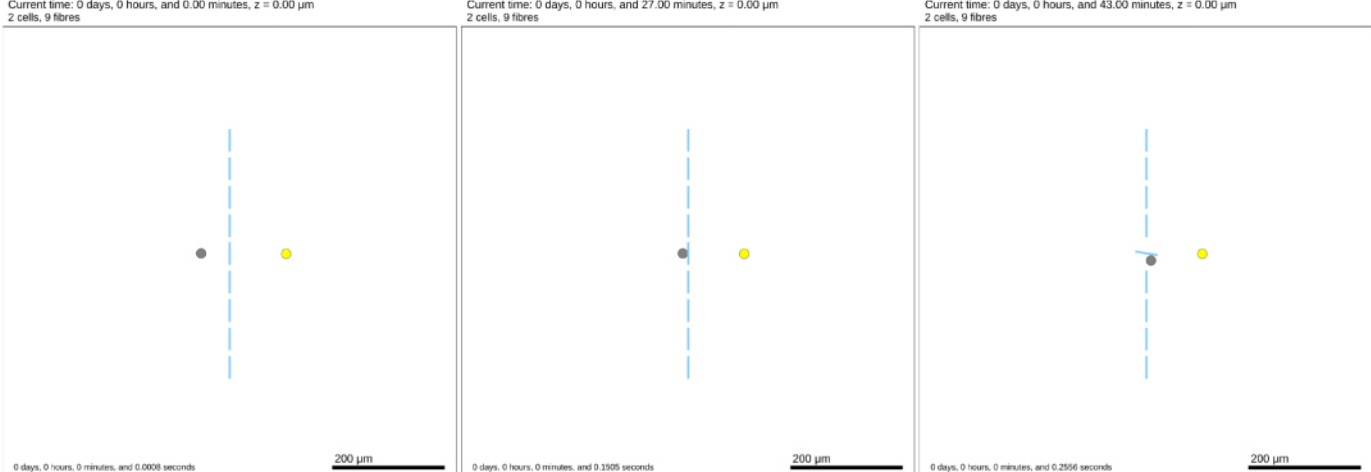

**Figure 5.** 2D Tissue domain with fibres forming a vertical barrier (blues lines), one chemotactic cell (grey) and one cell secreting an attractant (yellow) at time $t$ = 0 min (left), $t$ = 27 min (centre), $t$ = 43 min (right). At $t$ = 27 min, the chemotactic cell starts contact with the ECM fibre, then at $t$ = 43 min the cell was able to rotate the fibre to make a path to the attractant. Note: the time stated at the top of the figure is the simulated time and the time stated in the bottom right is the time taken to run the simulation.

**Angular velocity**:

$$\omega = \frac{J}{0.5 \cdot L^2}. \tag{5}$$

- $\omega$: Angular velocity. $J$: Impulse. $L$: Fibre length. This calculates the angular velocity induced by the impulse, inversely proportional to the fibre's moment of inertia (approximated here).

**New orientation**:

(i)  x-component:

$$o'_x = o_x \cdot \cos(\omega) - o_y \cdot \sin(\omega). \tag{6}$$

(ii)  y-component:

$$o'_y = o_x \cdot \sin(\omega) + o_y \cdot \cos(\omega). \tag{7}$$

- $o'_x$, $o'_y$: New orientation components. $o_x$, $o_y$: Old orientation components. $\omega$: Angular velocity. This set of equations updates the fibre's orientation based on the calculated angular velocity.

An example of fibre rotation is available in `Cell_Fibre_Mechanics/fibre_rotating.xml`, and can be visualised in Figure 5. The parameters controlling the cell–fibre mechanics are defined as custom data and expressed in the cell definitions of fibre-interacting cells, in the .xml file and detailed in Table 2.

Fibre-fibre interactions are fibre interactions with neighbouring fibres. For this release, we do not consider that there are fibre–fibre interactions. However, the method

**Table 2.** User parameter values for fibre mechanics read from .xml file.

| Equation Symbol | Parameter | Description |
| --- | --- | --- |
| $a_{\text{fibre}}$ | vel_adhesion | Adhesion coefficient for the additional adhesion between cells and fibres, parallel to fibre. |
| $\beta_{\text{fibre}}$ | vel_contact | Cell-fibre friction coefficient for the additional repulsion between cells and fibres, orthogonal to fibre. |
| $v_{\text{max}}$ | cell_velocity_max | Threshold velocity that limits the pulling effect fibres have on cells. |
| | fibre_pushing | Flag for fibre pushing by a cell (1 for active, 0 for inactive). |
| | fibre_rotation | Flag for fibre rotation by a cell (1 for active, 0 for inactive). |
| | fibre_sticky | Fibre friction - measure of how easily a cell can push or rotate a fibre. |

`add_potentials_from_fibre` is incorporated into the `PhysiMeSS_Fibre` class for future development.

### *Fibre degradation*

As discussed in the introduction to this paper, cellular remodelling of the ECM is a key process and an important interaction to consider between cells and matrix fibres. For this release of PhysiMeSS, we focused on matrix degradation. As well as degrading fibres, cells are responsible for creating new fibres and remodelling the ECM. The process of fibre generation by cells is not currently part of this release of PhysiMeSS but will be incorporated into a future release.

The degradation of the ECM plays a major role in tumour growth and invasion, for example. The key enzymes involved in this process are MMPs. MMPs can be found both within the ECM, at the cell membrane level (membrane-type MMPs), or being secreted by the cell itself. Different MMPs have different ECM-component targets, but their main function is to cleave and degrade structural ECM proteins such as elastin and collagen [21].

Fibre degradation is modelled in PhysiMeSS as a simple process of removing fibres from the domain under certain conditions. Fibre degradation can be turned on or off by the user using the Boolean flag `fibre_degradation`, a user parameter in the .xml file. During cell–fibre interactions as controlled by `add_potentials_from_fibre`, a fibre may be degraded if fibre degradation is turned on and further conditions are met, as detailed below. The fibre is then flagged for removal, which, for simplicity, happens instantly.

The first condition that must be met in order for a fibre to be degraded relates to whether a cell is hindered by the presence of a fibre. This may either be because as a cell tries to migrate through the domain, fibres act as obstacles in the cell's path, or it may be because as cells proliferate, they require additional space for the growing mass of cells. To target the first of these problems, the user can define a parameter `fibre_stuck_time` that determines how long a cell is stuck in a certain place before it will try to degrade/remove a fibre in its path. To determine whether a cell is stuck, we check as part of the function `physimess_update_cell_velocity` whether the magnitude of a cell's velocity is below a given threshold defined by the parameter `fibre_stuck_threshold`. In order that cells only remove fibres in their path, we consider the dot product of the cell's motility vector with the vector connecting the cell centre to the nearest point on the fiber; if this dot product is greater than zero, a fibre is said to be in a cell's path. A simple example of a cell migrating through the ECM and degrading cells along the way is available in `Fibre_Degradation/mymodel_fibre_degradation.xml`, and represented in Figure 6.

To target the second issue, whereby cells degrade fibres to make space for proliferation, we defined a new class in `custom.h`, called `PhysiMeSS_Fibre_Custom_Degrade`, which is



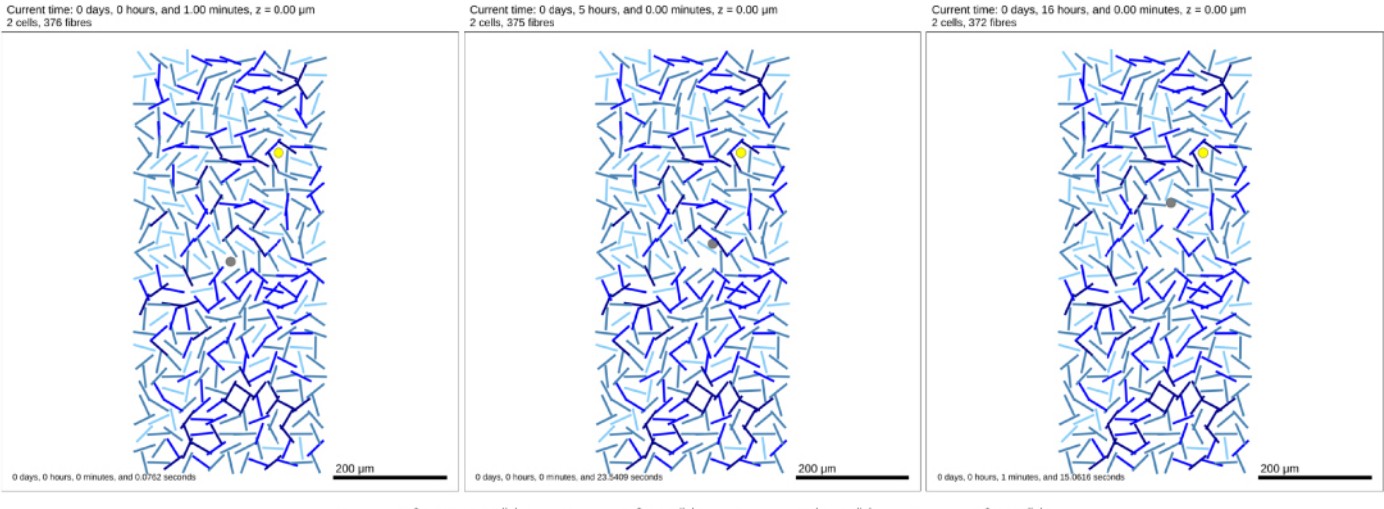

**Figure 6.** 2D Tissue domain with fibres (blues lines), one chemotactic cell (grey) and one cell secreting an attractant (yellow) at time $t$ = 1 min (left), $t$ = 5 h (centre), $t$ = 16 h (right). We can observe the chemotactic cell making its way to the attractant, degrading fibres on the way. Note: the time stated at the top of the figure is the simulated time and the time stated in the bottom right is the time taken to run the simulation.

**Table 3.** User parameter values for fibre degradation read from .xml file.

| Parameter | Description |
|---|---|
| `fibre_degradation` | Flag for fibre degradation (1 for active, 0 for inactive). |
| `fibre_stuck_time` | Number of mechanics timesteps a cell must be stuck before it can degrade a fibre at a rate `fibre_degradation_rate`. |
| `fibre_stuck_threshold` | Maximum movement size to declare a cell stuck. |
| `fibre_custom_degradation` | Flag for activating the degradation, which depends on pressure (1 for active, 0 for inactive). |
| `fibre_pressure_threshold` | Pressure threshold for cells above which they can degrade a fibre at a rate `fibre_degradation_rate`. |
| `color_cells_by_pressure` | Flag allowing the user to colour cells by the pressure they experience (1 for active, 0 for inactive). |
| `fibre_degradation_rate` | Rate of fibre degradation if fibre degradation is turned on via `fibre_degradation` and the cell is deemed to be stuck. |

derived from the `PhysiMeSS_Fibre` class. This new class implements in `custom.cpp` a custom degradation function, as described in Appendix D. In this custom implementation, we check whether the pressure experienced by a cell is above the threshold `fibre_pressure_threshold`. In order to indicate the pressure experienced by a cell, we have introduced an additional user Boolean flag that, when set to true, colours the cells by the pressure they experience using `color_cells_by_pressure`. If either of the constraints outlined are satisfied and fibre degradation is turned on, the cell will degrade neighbouring fibres at the rate `fibre_degradation_rate`. The custom implementation is controlled by a Boolean parameter, `fibre_custom_degradation`. This example is available in `Fibre_Degradation/mymodel_matrix_degradation.xml` and represented in Figure 7, where we compare degradations with and without this custom, pressure-dependent degradation.

All the parameters describing the fibre degradation by cells are controlled by custom data expressed in the cell definitions of fibre-interacting cells, in .xml file and detailed in Table 3.



**Figure 7.** 2D Tissue domain with fibres (blues lines), one initial cell coloured by the pressure sensed (blue: low, red: high) at time $t$ = 1 h (left), $t$ = 16 h (center), $t$ = 25 h (right). We show degradation independent on pressure (top), and dependent on pressure (bottom). We can observe the cell population growing, with high pressure when cells are trapped between fibres. Cells whose degradation rate responds to pressure are more able to reduce the pressure at the interface with fibres. At a later time in the pressure-dependent degradation condition, we obtain a gradient of high pressure in the middle of the population, lower at the border, as well as a rounder shape for the population, which is not observed in the case of the pressure-independant degradation. Note: the time stated at the top of the figure is the simulated time and the time stated in the bottom right is the time taken to run the simulation.

## CONCLUSIONS AND PLANNING FOR FUTURE RELEASES

In this paper, we outlined the features and capabilities of the inaugural release of PhysiMeSS, an add-on for PhysiCell. Here, we make the important note that PhysiMeSS has been designed as a framework to advance agent-based modelling in PhysiCell by introducing new cylindrical/rod-shaped agents. It is not designed to be the definitive model of how such agents behave; on the contrary, the user is encouraged to use custom functions suitable for their particular system of interest.

PhysiMeSS has been designed for modelling the ECM. However, we remind readers that the PhysiMeSS modelling framework goes beyond modelling tumour–ECM interactions, which originally sparked our interest, but can also model the ECM under other processes. Additionally, the new rod-shaped agents can be used to model non-ECM structures. As ECM

agents, they can interact with and be affected by cellular agents to permit detailed modelling of cell–fibre processes such as cell migration along and through the ECM network.

This release has focused on laying the groundwork for introducing these new agents and tackling the geometrical and topological aspects that then permit cell–fibre interactions. We have introduced some basic mechanics that may be useful when modelling cell–fibre interactions, but the user can add and expand to this using their own custom functions (Appendix D). One such possibility is to customise functions for the mechanics of fibre–fibre interactions. Fibre-fibre interactions are of high importance when it comes to modelling the dynamic network of fibres forming the ECM. In this release, there are no standard functions described for fibre–fibre interactions as this is intended to form future work. However, the PhysiMeSS framework has been designed so that the user could input their own mechanical interactions and model fibre–fibre interactions using this inaugural release of PhysiMeSS.

For the next release of PhysiMeSS, we will focus on two main aspects. Firstly, ensuring full 3D compatibility. At present, the potential inherent to the pushing and the rotation of fibres is limited to 2 dimensions. PhysiCell/PhysiMeSS gives the user the possibility to switch between a 2D and a 3D representation but does not take into account the modifications needed to rotate the fibres in 3D. Secondly, permitting fibrogenesis, i.e., the creation of new ECM fibres by cells.

## AVAILABILITY OF SOURCE CODE AND REQUIREMENTS

- Project name: PhysiMeSS
- Project home page: https://github.com/PhysiMeSS/PhysiMeSS
- Operating system(s): Platform independent
- Programming language: C++
- Other requirements: OpenMP
- License: BSD Clause 3.
- RRID:SCR_025455.

## DATA AVAILABILITY

Snapshots of the code are available in the GigaDB repository [22].

## Abbreviations

ABM, Agent-based model; CAF, cancer-associated fibroblast; ECM, extracellular matrix; FGFR1, fibroblast growth factor receptor-1; FN, fibronectin; MDE, matrix-degrading enzymes; MMP, matrix metalloproteinases.

## DECLARATIONS

### Ethics approval and consent to participate

The authors declare that ethical approval was not required for this type of research.

### Competing interests

The author(s) declare that they have no competing interests.

### Authors' contributions

VN, MR, RS, CKM all contributed to developing the software and writing the manuscript.

## Funding

VN and MR were supported by the European Commission under the PerMedCoE project [H2020-ICT-951773] and by the Inserm amorçage project. CKM gratefully acknowledges the support of her Rankin-Sneddon Fellowship at the University of Glasgow.

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

## APPENDIX A. NEW AGENTS CLASSES ADDED WITH THE PHYSIMESS ADDON

PhysiMeSS introduces new types of agents, derived from the `Cell` class that include more properties and behaviours. The first of them is `PhysiMeSS_Agent`, which directly inherits from `Cell`. The others are `PhysiMeSS_Fibre` and `PhysiMeSS_Cell`, which implement the fibres and the cells interacting with them, respectively, and directly inherit from `PhysiMeSS_Agent` (Figure 8).

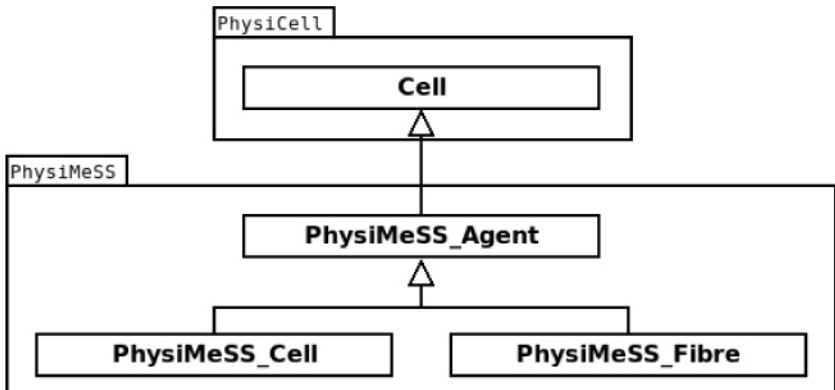

**Figure 8.** UML Class diagram showing the relations between `Cell` class and the classes introduced in PhysiMeSS.

`PhysiMeSS_Agent` adds two new properties: a list of *voxels* in which it is located, and a list of neighbours, which can include both cells and fibres. The list of *voxels* will be built using the `register_agent_voxels` method, which is defined as virtual, meaning that its implementation can be different in the derived classes. Both `PhysiMeSS_Cell` and `PhysiMeSS_Fibre` are implementing it, with the difference that cells only span one *voxel* while fibres can span multiple *voxels*. The list of neighbours is then built using the `find_agent_neighbors` method using the `find_agent_voxels` method, both implemented in `PhysiMeSS_Agent`.

`PhysiMeSS_Cell` adds new components to `PhysiMeSS_Agent` related to their interaction with fibres, such as the potential function `add_potentials_from_fibre`, which adds adhesion/repulsion forces. This method also calls the `degrade_fibre` method, which can trigger the degradation of a fibre by the cell. Both of these methods are declared virtual so that they can be overloaded if a user wants to modify their behaviour. Another aspect added by `PhysiMeSS_Cell` concerns what to do if a cell becomes trapped in ECM. Two variables, `stuck_counter` and `unstuck_counter`, are used to quantify this and potentially trigger the `force_updated_motility_vector` method to force the movement of the cell.

Finally, the `PhysiMeSS_Fibre` adds components necessary to define the fibre, as well as components related to the interaction with cells and other fibres. To define the fibre, we need to define its length (variable `fibre_Length`) and its radius (variable `fibre_Radius`). Its orientation is defined in the `Cell` variable `orientation`, which needs to be assigned via the `assign_fibre_orientation` method after the object is created. To deal with interaction with cells and fibres, we implemented the potential methods `add_potentials_from_fibre` and `add_potentials_from_cell`, which add forces from fibres or cells, respectively. Finally, to describe cross-links between fibres, we defined two variables, `fibres_crosslinkers` and `fibres_crosslink_points`, which are built using the method `check_fibre_crosslinks`.

## APPENDIX B. ADDING CYLINDRICAL FIBRES TO THE TISSUE SET-UP

For PhysiMeSS to identify agents as cylindrical matrix fibres, the user must give the cell definition a name containing any of the following recognised strings: {*ecm, fiber, fibre, matrix, rod*}. This will allow PhysiMeSS to properly recognise them using the `isFibre` function. Then, in the function `create_cell_types` in `custom.cpp`, users must assign a function instantiating the proper class to the `instantiate_cell` function pointer in `Cell_Functions`. That will allow those cell types to be properly instantiated with the corresponding class. Note that the same should also be done with standard cell types so that they can correctly interact with fibres. Fibres also need to have two specific function pointers assigned for correctly plotting them in the SVG: `plot_agent_SVG` and `plot_agent_legend`.

Fibres are initialised in the domain via the `setup_tissue` function in `custom.cpp`. If a .csv initial position file is provided, cells and fibres will be automatically created and `isFibreFromFile` will be turned to true. If there is no .csv file, the number of fibres initialised is given by `number_of_fibres` and the number of cells by `number_of_cells`. Parameters that govern the length, radius and orientation angle of fibres are read from the custom data of the fibre cell definition, in the .xml file (see Table 1). The length of fibres is normally distributed around a mean `fibre_length` with standard deviation `length_normdist_sd`. The radius of fibres is given by `fibre_radius`. The fibres will then have to be assigned an orientation via `assign_fibre_orientation`. Fibres can either be initialised isotropically or anisotropically within the domain controlled by the Boolean flag `anisotropic_fibres`. If `anisotropic_fibres` is false fibres are given a random orientation. If `anisotropic_fibres` is true the orientation of fibres is given as an angle from horizontal; this angle is normally distributed around a mean `fibre_angle` with standard deviation `angle_normdist_sd` and measured in radians. Finally, we need to check if fibres are out of bounds using `check_out_of_bounds`. Fibres determined to be out of bounds will be removed via the call to `remove_physimess_out_of_bounds_fibres`.

## APPENDIX C. GEOMETRICAL ISSUES: FIBRE NEIGHBOURS, FIBRE CROSS-LINKS AND FINDING THE NEAREST POINT ON A FIBRE

### Fibre neighbours

Since long cylindrical agents may exist in multiple *voxels* we must be able to determine each such *voxel* to correctly identify possible agent neighbours and interactions. The `PhysiMeSS_Agent` method `register_fibre_voxels` determines each *voxel* containing part of a fibre and registers the fibre to it. We start at one end of a fibre and sample *voxels* along the

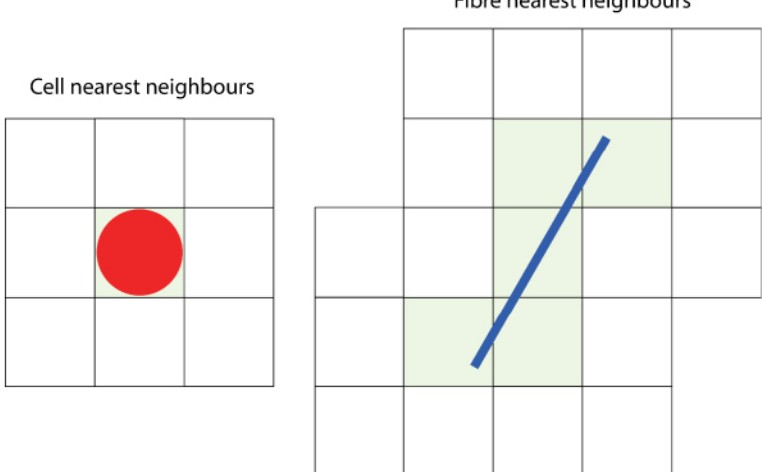

**Figure 9.** Agent voxels, for a cell (left) and for a fibre (right). These voxels will then be used to identify neighbours and interactions for each agent.

fibre at appropriate intervals no greater than the mechanics' *voxel* size determined by PhysiCell. A fibre is registered a maximum of once to each unique *voxel* found in this way.

The `PhysiMeSS_Agent` method `find_agent_voxels` creates a unique list of all the *voxels* any agent has been registered to and all adjacent *voxels*, and stores it in the `PhysiMeSS_Agent` variable `physimess_voxels`. If the agent is a fibre, this will be the multiple *voxels* determined by `register_fibre_voxels` and all *voxels* adjacent to any of those *voxels*. If the agent is a cell, this will be the single *voxel* containing its centre and any adjacent *voxels*. An example is shown in Figure 9. In each case, this function determines a unique list of *voxels* that could contain neighbour agents.

The `PhysiMeSS_Agent` method `find_agent_neighbors` determines if other agents sit within any of the *voxels* found in the list generated by `find_agent_voxels`. Any such agents are then declared neighbours of the the original agent and added to a list of neighbours defined in the `PhysiMeSS_Agent` variable `physimess_neighbors`.

The `PhysiMeSS_Fibre` method `deregister_fibre_voxels` takes all of the *voxels* a fibre has been registered to and de-registers it from all bar the *voxel* containing its centre. This method is called after the neighbours have been computed, to remove the fibre-specific voxels that would interfere with PhysiCell's code.

### Nearest point on fibre

The `PhysiMeSS_Fibre` method `nearest_point_on_fibre` determines for any given point and any given fibre the displacement vector connecting the point to the nearest point on the fibre, the orthogonal distance. This displacement vector can then determine the shortest distance between a point and a fibre to compare with interaction distances or is used alone to dictate the direction of forces between cells and fibres. The algorithm for determining the displacement vector from a fibre, *f*, to a point, *p*, is as follows:

- Select one endpoint of the fibre, $f_e$, and determine the vector that points from $f_e$ to *p*, i.e., $\vec{v} = \vec{p} - \vec{f_e}$.

- Determine the dot product between the vector, $\vec{v}$, and the direction vector of the fibre, $\vec{f}$, i.e., $\vec{v} \cdot \vec{f}$.
- If the dot product is less than zero, the nearest point on the fibre is the chosen endpoint and the displacement vector is $\vec{v}$.
- If the dot product is greater than the square of the fibre length, $|\vec{f}|$, then the nearest point on the fibre is the other end of the fibre and the displacement vector is $\vec{v} - \vec{f}$.
- If the dot product is greater than zero but less than the square of the fibre length, then the nearest point lies along the fibre. We calculate the distance, $l$, along the fibre from the original endpoint to this point by projecting $|\vec{v}|$ through an angle $\theta$, which is the angle between $\vec{v}$ and $\vec{f}$, and is easily calculated from $\cos\theta = (\vec{v} \cdot \vec{f})/(|\vec{v}||\vec{f}|)$. Thus, $l = (\vec{v} \cdot \vec{f})/|\vec{f}|$ and the displacement vector we require is $\vec{v} - l\hat{f}$, where $\hat{f}$ is the unit vector in the direction of the fibre.

## Checking fibre cross-links

The `PhysiMeSS_Fibre` method `check_fibre_crosslinks` checks if there are any intersections between two given fibres. If an intersection is detected, it will be added to the list of cross-linkers of both fibres. The function considers the spatial disposition of the two fibres verifying the intersection and proximity of the two fibres based on their geometric properties. Given a fibre $f_a$ and its neighbouring fibre $f_b$ (Figure 10), the possible intersection is determined as follows:

- Calculate fibre $f_a$ endpoints $f_{1e}, f_{2e}$ and fibre $f_b$ endpoints $f_{3e}$ and $f_{4e}$ using their positions, lengths, and orientations.
- Determine the direction vectors between the endpoints: from $f_{1e}$ to $f_{2e}$ ($\vec{v_{12}} = \vec{f_a}$), from $f_{3e}$ to $f_{4e}$ ($\vec{v_{34}} = \vec{f_b}$), from $f_{1e}$ to $f_{3e}$ ($\vec{v_{13}}$), and the vector between the centres of the fibres ($\vec{v_{cc}}$).
- Check if the fibres are co-planar and parallel by calculating the cross product of $\vec{v_{12}}$ and $\vec{v_{34}}$ ($\vec{f_{cp}}$). Verify that the dot product of $\vec{f_{cp}}$ with itself is zero, i.e., $\vec{f_{cp}} = \vec{v_{12}} \times \vec{v_{34}}$ and $\vec{f_{cp}} \cdot \vec{f_{cp}} = 0$.
  Condition 1: if the vector $\vec{v_{cc}}$ is parallel or anti-parallel to the current fibre orientation, and the centre to centre distance, $|\vec{v_{cc}}|$, is less than or equal to a reference length (equal to half the length of $f_a$ and half the length of $f_b$), then a cross-link is identified.
- Check if the neighbour fibre $f_b$ is stacked with $f_a$ so that fibres are not co-planar but parallel/anti-parallel. Check distances ($D_1$ and $D_2$) between fibre endpoints, $f_{1e}, f_{2e}$, respectively, and the point on fibre $f_b$ nearest to fibre $f_a$, taking advantage of the function `nearest_point_on_fibre`.
  Condition 2: if either the distance $D_1$ or $D_2$ is less than a reference length (equal to the sum of the radii of the two fibres) and the vector $\vec{v_{cc}}$ is not parallel or anti-parallel to the current fibre orientation, then a cross-link is identified.
- Check if the fibres are co-planar, skewed, and intersecting. This typically requires the scalar triple product to be zero. For skewed fibres that intersect, compare the scalar triple product (between $\vec{v_{12}}$ and $\vec{f_{cp}}$) to a reference value equal to the sum of the radii. To determine if the skewed fibres intersect, use the line equations for each fibre: $L_a: f_{1e} + t_1(f_{2e} - f_{1e})$; $L_b: f_{3e} + t_2(f_{4e} - f_{3e})$, and solve for $t_1$ and $t_2$.
  Condition 3: if $t_1$ and $t_2$ both lie within [0,1], allowing for a tolerance (equal to the sum of the radii of the two fibres normalised by the sum of the two fibres half-lengths), then a cross-link is identified.



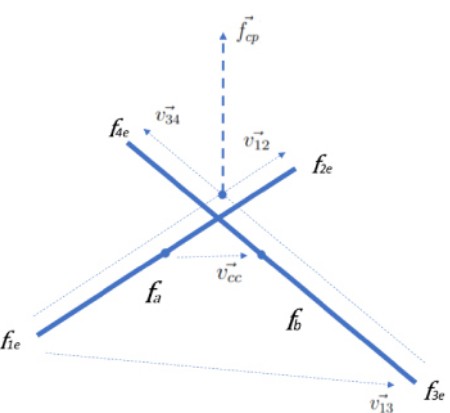

**Figure 10.** Schema of intersecting fibres.

## APPENDIX D. USER CUSTOMISATION OF PHYSIMESS

The user can modify much of what is documented and described in this paper simply by changing user parameters (Tables 1, 2 and 3). However, they may also wish to include their own (mechanical) interactions between cells and fibres.

To do this, PhysiMeSS allows users to create a new class to implement new variations of the potential and degradation functions, derived from `PhysiMeSS_Fibre` and `PhysiMeSS_Cell`. The `PhysiMeSS_Cell` method `add_potentials_from_fibre` and the `PhysiMeSS_Fibre` methods `degrade_fibre`, `add_potentials_from_cell`, and `add_potentials_from_fibre` are all declared as virtual methods, allowing derived classes to overload them. Users then need to create functions instantiating these newly derived classes and select them in the `instantiate_cell` function pointer of the desired cell definition.

To show such a possibility we added an example of a new degradation function using pressure exerted on cells to increase the probability of degradation, used in the example `Fibre_Degradation/mymodel_matrix_degradation.xml`. We start by declaring a new class `PhysiMeSS_Cell_Custom_Degrade` in `custom.h`, which inherits from `PhysiMeSS_Cell`, and declaring one method `degrade_fibre`, which will overload the one in `PhysiMeSS_Cell`.

```
class PhysiMeSS_Cell_Custom_Degrade : public PhysiMeSS_Cell
{
  public:
  void degrade_fibre(PhysiMeSS_Fibre* pFibre);
};
```

Then in `custom.cpp` we define this new method:

```
void PhysiMeSS_Cell_Custom_Degrade::degrade_fibre(PhysiMeSS_Fibre* pFibre)
{
  ...
}
```

Then we define a new function to instantiate this new object:

```
Cell* instantiate_physimess_cell_custom_degrade() { return new PhysiMeSS_Cell_Custom_Degrade; }
```

that we finally use in the `instantiate_cell` function pointer:

```
cell_defaults.functions.instantiate_cell = instantiate_physimess_cell_custom_degrade;
```

The same construct can be applied with potential functions, allowing the advanced user to modify the mechanical interactions between cells and fibres.


