## [Editor Report]

Editor’s AssessmentPhysiCell is an open source multicellular systems simulator for studying many interacting cells in dynamic tissue microenvironments. As part of the PhysiCell ecosystem of tools and modules this paper presents a PhysiCell addon, PhysiMeSS (MicroEnvironment Structures Simulation) which allows the user to accurately represent the extracellular matrix (ECM) as a network of fibres. This can specify rod-shaped microenvironment elements such as the matrix fibres (e.g. collagen) of the ECM, allowing the PhysiCell user the ability to investigate physical interactions with cells and other fibres. Reviewers asked for additional clarification on a number of features. And the paper now clear future releases will provide full 3D compatibility and include working on fibrogenesis, i.e. the creation of new ECM fibres by cells.Editor’s AssessmentPhysiCell is an open source multicellular systems simulator for studying many interacting cells in dynamic tissue microenvironments. As part of the PhysiCell ecosystem of tools and modules this paper presents a PhysiCell addon, PhysiMeSS (MicroEnvironment Structures Simulation) which allows the user to accurately represent the extracellular matrix (ECM) as a network of fibres. This can specify rod-shaped microenvironment elements such as the matrix fibres (e.g. collagen) of the ECM, allowing the PhysiCell user the ability to investigate physical interactions with cells and other fibres. Reviewers asked for additional clarification on a number of features. And the paper now clear future releases will provide full 3D compatibility and include working on fibrogenesis, i.e. the creation of new ECM fibres by cells.

---

## [Reviewer Report]

Reviewer name and names of any other individual's who aided in reviewerErika TsingosDo you understand and agree to our policy of having open and named reviews, and having your review included with the published manuscript. (If no, please inform the editor that you cannot review this manuscript.)YesIs the language of sufficient quality?YesPlease add additional comments on language quality to clarify if neededIs there a clear statement of need explaining what problems the software is designed to solve and who the target audience is? YesAdditional CommentsIs the source code available, and has an appropriate Open Source Initiative license <a href="https://opensource.org/licenses" target="_blank">(https://opensource.org/licenses)</a> been assigned to the code?YesAdditional CommentsAs Open Source Software are there guidelines on how to contribute, report issues or seek support on the code?YesAdditional CommentsIs the code executable?YesAdditional CommentsIs installation/deployment sufficiently outlined in the paper and documentation, and does it proceed as outlined?YesAdditional CommentsIs the documentation provided clear and user friendly?YesAdditional CommentsIs there enough clear information in the documentation to install, run and test this tool, including information on where to seek help if required?YesAdditional CommentsIs there a clearly-stated list of dependencies, and is the core functionality of the software documented to a satisfactory level?YesAdditional CommentsHave any claims of performance been sufficiently tested and compared to other commonly-used packages? Not applicableAdditional CommentsIs test data available, either included with the submission or openly available via cited third party sources (e.g. accession numbers, data DOIs)?YesAdditional CommentsAre there (ideally real world) examples demonstrating use of the software? YesAdditional CommentsIs automated testing used or are there manual steps described so that the functionality of the software can be verified?YesAdditional CommentsAny Additional Overall Comments to the AuthorOne important aspect that the authors need to be aware of and mention explicitly is that their algorithm for fiber set-up leads to differences in fiber concentration and orientation at the boundary, because fibers that are not wholly contained in the simulation box are discarded. The effect of this choice can be seen upon close inspection of Figure 2: In the left panel, fibers align tangentially to the boundary, so locally the orientation is not isotropic. Similarly, in Figure 2 middle and right panels, the left and right boundaries have lower local fiber concentration. This issue could potentially affect the outcome of a simulation, so it's important that readers are made aware so that if necessary they can address this with a modified algorithm. ----- Minor comments: In the abstract, the phrasing implies agent-based frameworks are only used for tumour evolution. I would rephrase such that it is clear that tumour evolution is one example among many possible applications. I suggest adding a dash to improve readability in the following sentence in the introduction:
"However, we note that the applications of PhysiMeSS stretch beyond those wanting to model the ECM -- as the new cylindrical/rod-shaped agents could be used to model blood vessel segments or indeed create obstacles within the domain." In the implementation section, add a short sentence to clarify if PhysiMeSS is "backwards compatible" with older PhysiCell models that do not use the fiber agent. Notation in equations: A single vertical line is absolute value, and two vertical lines is Euclidean norm? The explanation of Equation 1 implies that the threshold v_{max} should limit the parallel force, but the text does not explicitly say if ||v|| is restricted to be less or equal to v_{max}. Is that the case? In Equation 2, I don't see the need to square the terms in parenthesis. If |v*l_f| is an absolute value it is always positive. Since l_f is normalized the value of the dot product is only between 0 and the magnitude of v. Am I missing something? Are p_x and p_y in the moment arm magnitude coordinates with respect to the fiber center? Table 2: It would be helpful to have a separate column with the corresponding symbols used throughout the text and equations. Figure 5/6: Missing crosslinker color legend. ----- Typos/grammar:
"As an aside, an not surprisingly,"  As an aside, and not surprisingly,
"This may either be because as a cell tries to migrate through the domain fibres which act as obstacles in the cell’s path,"  remove the word "which"RecommendationMinor Revisions

---

## [Reviewer Report]

Reviewer name and names of any other individual's who aided in reviewerJinSeok ParkDo you understand and agree to our policy of having open and named reviews, and having your review included with the published manuscript. (If no, please inform the editor that you cannot review this manuscript.)YesIs the language of sufficient quality?YesPlease add additional comments on language quality to clarify if neededIs there a clear statement of need explaining what problems the software is designed to solve and who the target audience is? YesAdditional CommentsIs the source code available, and has an appropriate Open Source Initiative license <a href="https://opensource.org/licenses" target="_blank">(https://opensource.org/licenses)</a> been assigned to the code?YesAdditional CommentsAs Open Source Software are there guidelines on how to contribute, report issues or seek support on the code?YesAdditional CommentsIs the code executable?YesAdditional CommentsIs installation/deployment sufficiently outlined in the paper and documentation, and does it proceed as outlined?YesAdditional CommentsIs the documentation provided clear and user friendly?YesAdditional CommentsIs there enough clear information in the documentation to install, run and test this tool, including information on where to seek help if required?YesAdditional CommentsIs there a clearly-stated list of dependencies, and is the core functionality of the software documented to a satisfactory level?YesAdditional CommentsHave any claims of performance been sufficiently tested and compared to other commonly-used packages? YesAdditional CommentsIs test data available, either included with the submission or openly available via cited third party sources (e.g. accession numbers, data DOIs)?YesAdditional CommentsAre there (ideally real world) examples demonstrating use of the software? YesAdditional CommentsIs automated testing used or are there manual steps described so that the functionality of the software can be verified?Additional CommentsAny Additional Overall Comments to the AuthorNoel et al. introduce PhysiMess - a new PhysiCell Addon for ECM remodeling. This new addon is a powerful tool to simulate ECM remodeling and has the potential to be applied to mechanobiology research, which makes my enthusiasm high. I would like to give a few suggestions. 1) Basically, it is an addon of PhysiCell. So, I suggest describing PhysiCell and how to add the addon for readers who are not familiar with these tools. Also, screen captures of tool manipulation would be very helpful. 2) Figure 2 and 3 exhibit the outcome of the addon showing ECM remodeling. I would suggest to show actual ECM images modeled by the addon. 3) The equations reflect four interactions, and in my understanding, the authors describe cell-fibre, fiber-cell, and fiber-fiber interactions. I suggest generating an example corresponding to each interaction's modulation and explaining how the add-on results explain the physiological phenomena. For instance, focal adhesion may be a key modulator of cell-fibre or fiber-cell interaction, presumably, alpha or beta fiber. I would demonstrate how the different parameters generate different results and explain the physiological situation modeled by the results. 4) Similarly, Figure 5 and Figure 6 only show one example and no comparison with other conditions. For example, It would be better to exhibit no pressure/pressure conditions. It may help readers estimate how the pressure impacts cell proliferation.RecommendationMinor Revisions

---

## [Reviewer Report]

Upload additional filesTRR-202407-01R01/stage_files/TRR-202407-01/Review MS/gx-TR-1719946069_SY.pdfReviewer name and names of any other individual's who aided in reviewerSimon SygaDo you understand and agree to our policy of having open and named reviews, and having your review included with the published manuscript. (If no, please inform the editor that you cannot review this manuscript.)YesIs the language of sufficient quality?YesPlease add additional comments on language quality to clarify if neededIs there a clear statement of need explaining what problems the software is designed to solve and who the target audience is? YesAdditional CommentsIs the source code available, and has an appropriate Open Source Initiative license <a href="https://opensource.org/licenses" target="_blank">(https://opensource.org/licenses)</a> been assigned to the code?YesAdditional CommentsAs Open Source Software are there guidelines on how to contribute, report issues or seek support on the code?YesAdditional CommentsIs the code executable?Unable to testAdditional CommentsIs installation/deployment sufficiently outlined in the paper and documentation, and does it proceed as outlined?YesAdditional CommentsIs the documentation provided clear and user friendly?YesAdditional CommentsIs there enough clear information in the documentation to install, run and test this tool, including information on where to seek help if required?YesAdditional CommentsIs there a clearly-stated list of dependencies, and is the core functionality of the software documented to a satisfactory level?YesAdditional CommentsHave any claims of performance been sufficiently tested and compared to other commonly-used packages? Not applicableAdditional CommentsIs test data available, either included with the submission or openly available via cited third party sources (e.g. accession numbers, data DOIs)?YesAdditional CommentsAre there (ideally real world) examples demonstrating use of the software? YesAdditional CommentsWill be used in scientific simulations.Is automated testing used or are there manual steps described so that the functionality of the software can be verified?NoAdditional CommentsThere seem to be no dedicated tests for the PhysiMeSS package extension.Any Additional Overall Comments to the AuthorThe presented paper "PhysiMeSS - A New PhysiCell Addon for Extracellular Matrix Modelling" is a useful extension to the popular simulation framework PhysiCell. It enables the simulation of cell populations interacting with the extracellular matrix, which is represented by a set of line segments (2D) or cylinders (3D). These represend a new kind of agent in the simulation framework. The paper outlines the basic implementation, properties and interactions of these agents. I recommend publication after a small set of minor issues have been addressed. Please refer to the attached marked-up PDF file for these minor issues and suggestions.RecommendationMinor Revisions